

# Best in the company of nearby males: female success in the threatened cycad, *Zamia portoricensis*

Julio C. Lazcano-Lara and James D. Ackerman

Department of Biology, University of Puerto Rico, San Juan, Puerto Rico

## ABSTRACT

Variation in plant reproductive success is affected by ecological conditions including the proximity of potential mates. We address the hypothesis that spatial distribution of sexes affects female reproductive success (RS) in the dioecious cycad, *Zamia portoricensis*. Are the frequencies of males, operational sex ratios, and distances to the nearest mate associated with RS in females? We studied the spatial distribution of sexes in two populations in Puerto Rico and compared RS of target females with the number of males and operational sex ratios. Population structure suggests regular successful recruitment. Adults, males, and females were randomly distributed with respect to one another. Reproductive success of females was highly variable, but was higher in neighborhoods with more males than females and generally decreased with increasing distance to the nearest male, becoming statistically significant beyond 190 cm. This possible mate-finding Allee effect indicates that pollinator movement among plants may be limited for this mutually dependent plant-pollinator interaction. Yet being close to male plants is a matter of chance, perhaps a factor generating the high intra-population genetic diversity in *Z. portoricensis*.

Corresponding author
Julio C. Lazcano-Lara,
jlazcano1@yahoo.com

## INTRODUCTION

Sexual systems (e.g., dioecy or monoecy), mating systems (e.g., selfing or outcrossing), and pollination systems (e.g., generalist or specialist vector, reward or deception mediated pollination) influence the dispersal of alleles and play a major role in the evolution of plants (*Barrett, 2003*; *Barrett, 2013*; *Devaux, Lepers & Porcher, 2014*). Mating systems strongly influence the patterns of genetic structure of populations. In fact, outcrossing and selfing leave a distinctive imprint on the genetic constitution of species, which is particularly notable in their levels of genetic diversity (*Hamilton, 2009*). Obligate outcrossing species (e.g., dioecious species) are particularly affected by access to mates, which is influenced by the pollination system and the spatial distribution of individuals (*Barrett & Thomson, 1982*; *Heilbuth, Ilves & Otto, 2001*; *Ghazoul, 2005*) resulting in limitations to the reproductive success that may have constrained the spread of dioecy among plants (*Barrett, 2013*).

Reproductive success in obligately outcrossing plant species are often negatively correlated with distance to the nearest mate and positively correlated with population

size (*Barrett & Thomson, 1982*; *Percy & Cronk, 1997*; *Metcalfe & Kunin, 2006*; *Gascoigne et al., 2009*). These are mate-finding Allee effects, which can be most prevalent in dioecious species where both population size and sex ratios can affect reproductive success (*Cabral & Schurr, 2010*; *Xia et al., 2013*). Female advantage in the presence of more males is not only observed empirically, but also by modeling studies (*Gascoigne et al., 2009*). Cycads, an ancient gymnosperm lineage and perhaps the first seed plants to be insect pollinated, may be particularly susceptible to Allee effects given that they are dioecious, and many species occur as scattered, low density, populations (*Donaldson, 2003*). Here we address the hypothesis that spatial distribution of sexes within populations affects female reproductive success even in relatively dense natural populations of *Zamia portoricensis* Urb. in Puerto Rico.

We sought to determine whether variation in spatial relationships was an indicator of success in sexual reproduction. Specifically, we asked (1) whether sexes of *Z. portoricensis* are randomly distributed within populations; 2) is the abundance of coning males associated with reproductive success (RS), and (3) how does RS relate to distance to the nearest potential mate?

## MATERIALS AND METHODS

### Study species

*Z. portoricensis* are short plants with thick, bending upright, simple to sparsely branched, underground stems. It is a perennial, dioecious gymnosperm endemic to the south-central and southwestern serpentine outcrops and limestone zones of Puerto Rico (*Acevedo-Rodríguez & Strong, 2005*). The reproductive cycle of the species generally starts late in October with the appearance of the first male cones; female cones become receptive from January to February and, if pollinated, they will support maturing seeds until cone dehiscence around November. Ovules are pollinated by *Zamia*-specialist beetles, *Pharaxonota portophylla* (Coleoptera: Erotylidae) (*Franz & Skelley, 2008*). This interaction involves a sequential brood-site reward-deception pollination system, where the plants and the insects are mutually dependent upon one another (*Norstog, Fawcett & Vovides, 1992*). Beetles exclusively feed on pollen and male cone tissue, probably mate inside the cones, and use them as a brood place (*Norstog & Nicholls, 1997*). The life span of the insects is unknown, but several generations are produced during the host reproductive season (*Norstog, Fawcett & Vovides, 1992*; *Norstog & Nicholls, 1997*). Pollination occurs when receptive female plants produce volatile compounds that mimic male scents and attract pollen-covered insects (*Pellmyr et al., 1991*; *Terry et al., 2004*; *Proches & Johnson, 2009*). Ovular micropyles produce drops rich in sugars and amino acids. The drops are not likely a reward because they are produced in small quantities when insects are not active inside female cones (*Tang, 1987a*; *Tang, 1993*). Although the evidence is not conclusive, thus far female plants can be considered rewardless (*Donaldson, 1997*). As other zamias of the West Indies, *Z. portoricensis* experiences short seed dispersal distances (*Eckenwalder, 1980*; *Negrón-Ortiz & Breckon, 1989a*; *Tang, 1989*), and seed production may or may not be pollen limited (*Newell, 1983*; *Tang, 1987a*). Despite having seeds with a fleshy,

brightly colored outermost layer, effective animal dispersal has only been scientifically, not anecdotally, documented in a very small number of cycad species (*Burbidge & Whelan, 1982*; *Ballardie & Whelan, 1986*; *Tang, 1989*; *Hall & Walter, 2013*). All Puerto Rican zamias (i.e., *Z. erosa*, *Z. portoricencis* and *Z. pumila*) have brightly red/pink/orange colored seeds, but no dispersal agent has been unequivocally identified for them (*Negrón-Ortiz & Breckon, 1989a*). In the most recent assessment of its conservation status, based on IUCN Red List Categories and Criteria, *Z. portoricensis* is considered Endangered (EN) (*Stevenson, 2010*).

## Study sites

Two populations of *Z. portoricensis*, both established on the serpentine outcrop of southern Puerto Rico, were included in this study. One population is located in the Municipality of Sabana Grande on both sides of El Tamarindo Road (Km 2.8 from intersection with Road 368), and the other population is in the Municipality of Yauco, in the Susúa State Forest, along the dirt road between the main campground and the cabins. Our permit numbers from Puerto Rico Department of Natural and Environmental Resources are 2012-IC-028, 2013-IC-022, 2014-IC-033. The zamias in El Tamarindo, hereafter named the ET population, are the most abundant element in the understory of a low (<6 m), partially open, semi-deciduous forest growing on a steep slope, where leaf litter is periodically flushed out by heavy rains (Fig. 1A). Canopy cover was measured with a spherical densiometer and plot averages were $63 \pm 32\%$ and $72 \pm 16\%$. The zamias in Susúa State Forest, hereafter named the SF population, are also the most abundant element in the understory of a high (>12 m), closed canopy, semideciduous forest. Canopy coverage averaged $93 \pm 2\%$ and $94 \pm 2\%$. The zamias were growing on nearly flat terrain at SF, with soil that periodically accumulates organic matter (Fig. 1B). In both sites, *Z. portoricensis* has a continuous distribution along the northern slopes of the hills. Species density is variable, sometimes appearing as isolated individuals, elsewhere forming dense clusters of plants, particularly at mid slope.

## Sampling

We established four $25 \times 20$ m plots, two per study site, to characterize both populations. In each plot, we recorded plant size, measured as the number of leaflets of the longest leaf, as previous studies on another Puerto Rican zamia, *Z. erosa*, showed that it is a good estimator to group individuals into size/age classes (*Negrón-Ortiz & Breckon, 1989b*; *Negrón-Ortiz, Gorchov & Breckon, 1996*) and, when possible (i.e., when they produced cones or when remnants of previous cones were present), the sex of every reproductive individual. To describe the population structure, we used the minimum size at which plants produced a cone to be the division line between juvenile and adult size classes. Within the adult class, coning plants were assigned to each sex, and the rest was counted as non-reproductive adults. Both populations were regularly visited for three consecutive years from January 2012 to December 2014; thus, the data were frequently updated. Sex ratio bias was evaluated using Chi-squared Goodness-of-Fit test, and differences in plant size among sites, plots, and sex were tested using a nested ANOVA. Normality of the data was corroborated using

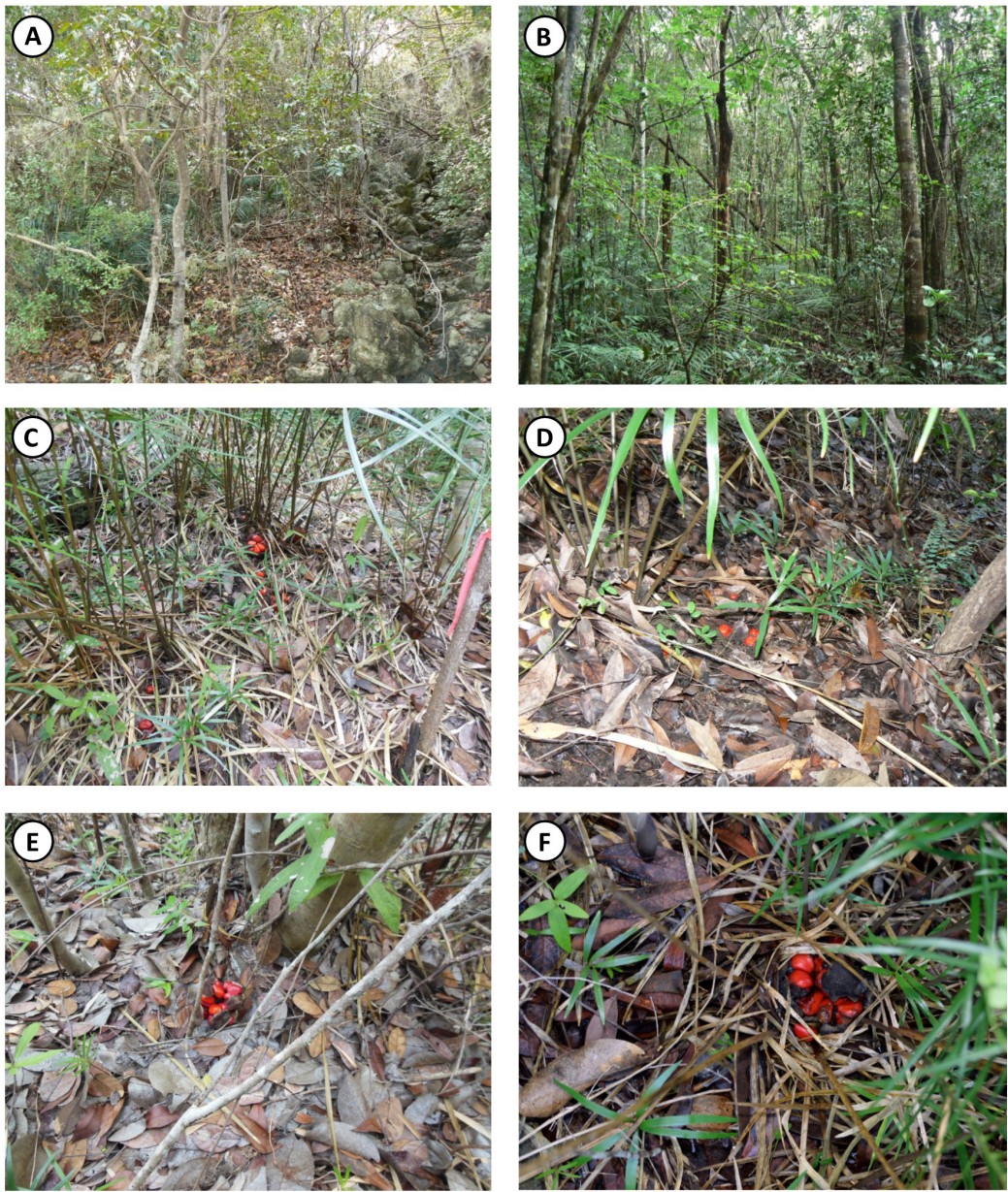

**Figure 1** **Habitat physiognomy and examples of limited dispersal of *Z. portoricensis* in the southcentral serpentine outcrop of Puerto Rico.** (A) El Tamarindo site, showing the relatively open, low canopy on steep slopes; (B) Susúa State Forest site, showing the closed, high canopy on relatively gentle slopes; (C–F) four examples of limited dispersal in *Z. portoricensis*; (C) seeds individually dispersed on a flat surface with no particular direction; (D) seeds individually dispersed on steep surface with downhill direction; (E) seeds dispersed as a group, in a cone fragment; (F) seeds remained as a group in a rotten but unbroken cone, at the mother plant. All photographs taken by Julio C. Lazcano Lara.

diagnostic graphics and residual analysis. Analyses were performed using R software (*R Development Core Team, 2014*) except the nested ANOVA which was carried out in Statistica (*StatSoft, Inc, 2004*).

## Spatial patterns and interaction among individuals

To characterize the spatial arrangement of the individuals of *Z. portoricensis*, we first recorded the geographic position of all mapped juveniles and adults. The spatial data were processed using Spatstat, an R package for analyzing point patterns in two dimensions (*Baddeley & Turner, 2005*; *Baddeley, 2010*). Limited local dispersal of the West Indies zamias (*Eckenwalder, 1980*; *Negrón-Ortiz & Breckon, 1989a*; *Tang, 1989*) (Figs. 1C–1F), originates a non-stationary (i.e., inhomogeneous) point process. Thus, to account for this lack of homogeneity, an adjustment was applied to analyze the interaction between points in the patterns obtained for each plot (*Baddeley, Moller & Waagepetersen, 2000*; *Perry, Miller & Enright, 2006*; *Law et al., 2009*; *Baddeley et al., 2010*). For each plot, we analyzed three sets of individuals: all individuals mapped; adults (reproductive and non-reproductive) and reproductive adults (females and males). We used the Linhom ($r$) function, which calculates an estimate of the inhomogeneous version of $L$-function (Besag's transformation of Ripley's K-function). For the distance argument ($r$) we used the range (0, $r$ max) recommended by Spatstat (0, 50 units, 1 unit = 0.1m). Rejection limits were estimated as simulation envelopes based on a null hypothesis of Complete Spatial Randomness (CSR) (*Perry, Miller & Enright, 2006*; *Baddeley, 2010*). We used 499 replicates of a Monte Carlo simulation to obtain the significance envelopes for $\alpha = 0.01$ (*Perry, Miller & Enright, 2006*).

## Reproductive effort

We expected males and females to initiate coning at different sizes given the expected difference between the sexes in the cost of reproduction and in some physiological traits (*Clark & Clark, 1987*; *Clark & Clark, 1988*; *Tang, 1990*; *Krieg et al., 2017*). Minimum plant size, as determined by the number of leaflets on the largest leaf, was assessed for the initiation of coning for male and female plants in each of the plots as described above. We also expected the size of female cones would be dependent on plant size. Leaflet data were normally distributed whereas ovule data were square root transformed to achieve normality (Shapiro–Wilks normality test). We then ran a linear regression using JMP® v. 5.1 (*SAS Institute, Inc, 2002*).

## Reproductive success

In March 2013, we created 60 non-overlapping circular plots of 1.5 m radius in and around the areas used for spatial analyses. At the center of each plot was a coning female plant, which we assessed for reproductive effort and success. In October 2013, approximately seven months after pollination took place and one month before seeds were completely ripe, the female cones were enclosed in perforated plastic bags to prevent seed loss from shedding mature cones, and by the end of November 2013, we recorded the total number of ovules produced per cone, and the number of seeds set. Only one plant produced more than a single cone and for this plant we averaged the data. Reproductive success was measured as seed production and also the percentage of ovules that developed into

seeds (i.e., seed set), both of which reflect the relationship between reproductive effort and pollination success.

To determine the relationship between seed set and pollen availability we did a linear regression analysis of reproductive success on the number of males, and the sex ratio (i.e., males/females). In each circular plot, male and female plants were counted. Pollen availability was assumed to increase with an increase in the male-to-female sex ratio. Normality of the data was tested using Shapiro–Wilks test. Square root transformations were done on explanatory variables: number of males, sex ratios, and distance to nearest male. The analyses were performed using JMP$^{®}$ v. 5.1 (*SAS Institute, Inc, 2002*). We also tested for spatial autocorrelation between plant locations within each of the four subplots and RS of target females using Mantel statistics based on Pearson's product-moment correlation. We used the community ecology statistical package (vegan) in R software (*R Development Core Team, 2014*).

The relationship between seed set and distance to the nearest male was assessed through a linear regression analysis. We included 85 plants in this study (i.e., 44 from El Tamarindo and 41 from Susúa State Forest). Before combining the data from both sites, we analyzed them separately and found their slopes to be statistically similar (i.e., ET: $b_{ET} = -0.19$, $r^2 = 0.11$, $P < 0.05$; SF: $b_{SF} = -0.33$, $r^2 = 0.27$, $P < 0.001$; H$_0$: $b_{ET} = b_{SF}$, $Z = 1.26$, $P = 0.21$). To identify a threshold distance after which the effect of the separation between a female plant and the nearest male results in a significant reduction of reproductive success we used an approach similar to the one used by *Ackerman, Trejo-Torres & Crespo-Chuy (2007)*. We ran a linear regression between reproductive success and distance to the nearest male. We then repeatedly re-ran the regression by progressively eliminating the next longest distance until the relationship became unambiguously non-significant ($P > 0.1$). We considered that last distance as the approximate threshold. Linear regressions were performed using R software (*R Development Core Team, 2014*).

## Patterns of female success

While seed production in *Zamia* is pollinator-dependent, it may or may not be pollen limited (*Newell, 1983*; *Tang, 1987a*). We sought indirect evidence of pollen or resource constraints using three approaches. For the first, we used a linear regression on seed set (proportion of ovules setting seed) as the response variable with female cone size (the number of ovules produced) as the explanatory variable. If plants are pollen limited, then there should be a negative relationship between the two (*Montalvo & Ackerman, 1987*). A positive relationship could occur if larger cones attract proportionately greater number of pollinators, have greater seed set, and if resources are not limiting. While zamias may not show a relationship between female cone size and plant size, they do show a cost to female reproduction (*Newell, 1983*; *Clark & Clark, 1988*). Lack of relationship suggests that resource availability is proportionate to the size of the cone such that seed set remains the same regardless of increases in pollen availability (e.g., *González-Díaz & Ackerman, 1988*).

In the second approach we asked whether a relationship exists between seed production (number of seeds produced in a cone) and female cone size because seed number and seed set (or fruit number and fruit set in the case of angiosperms) need not co-vary

**Table 1** Summary of reproductive and population structure data of the plants inventoried at two populations of *Z. portoricensis*.

| Plot | Plants | Density[a] | M | F | M/F[b] | SSJ | LJ | NonRA | %RA |
|------|--------|---------|-----|-----|---------|-------|-----|-------|-----|
| ET1 | 3,099 | 6.2 | 250 | 187 | 1.34* | 2,100 | 220 | 342 | 44 |
| ET2 | 612 | 1.2 | 59 | 46 | 1.28 | 410 | 22 | 75 | 52 |
| SF1 | 1,065 | 2.1 | 73 | 34 | 2.14** | 804 | 43 | 111 | 41 |
| SF2 | 913 | 1.8 | 83 | 24 | 3.46*** | 258 | 194 | 354 | 16 |

**Notes.**

ET, El Tamarindo population plots; SF, Susúa State Forest plots; M, males; F, females; SSJ, Seedlings and small juveniles; LJ, large juveniles; RA, reproductive adults.

[a] Plants per $m^2$ includes all life history stages.

[b] Sex ratio and significant Chi-square Goodness of Fit test results for sex ratio deviation from 1:1.

*$P = 0.003$.

**$P < 0.001$.

***$P < 0.0001$.

(*Montalvo & Ackerman, 1987*). Pollen limitation may arise or become exacerbated when larger cones do not attract more pollinators, so that pollen availability remains constant and additional ovules do not result in more seeds. A positive relationship between seed number and cone size would indicate that pollen is not limiting, and/or the larger cones attract more pollinators.

Finally, we surmised that resource competition would be greater in circular plots with more adult plants, particularly since plants often occur in high densities at our sites. We expected to see smaller cones (fewer ovules produced) in female plants occupying the more crowded plots than those more sparsely populated. We tested this with a Spearman's rank correlation in JMP (*SAS Institute, Inc, 2002*).

# RESULTS

## Population structure

Abundance of *Z. portoricensis* was variable across our study sites. We inventoried 5,689 plants (Table 1, Data S1–S4) and found densities of reproductive adults, to be $0.9/m^2$ in ET1, and $0.2/m^2$ in all other plots. Despite the difference, *Z. portoricensis* was the prominent understory element in all plots (Figs. 1A, 1B).

The percentage of adult plants, inferred from size class data, ranged from 20% (plot SF1) to 50% (plot SF2). The number of reproductively active adults (i.e., plants that actually produced cones within the adult size class) ranged from 16% (plot SF2) to 52% (plot ET2) of the plants inventoried (Table 1).

Operational sex ratios were male biased and significantly so in three of the four plots (Table 1). Since the sexual identity could not be determined in non-reproductive adults, the actual sex ratio of these populations is not known.

## Spatial patterns and interactions between individuals

The analysis of each set of spatial point pattern revealed that, except for the plot ET2, individuals are randomly distributed (Figs. 2A–2C, 2G–2L; Fig. S1). At ET2, the spatial pattern of two sets, all individuals mapped and all the adults, varied through the distance range ($r$). Those sets showed significant deviations from complete spatial randomness at

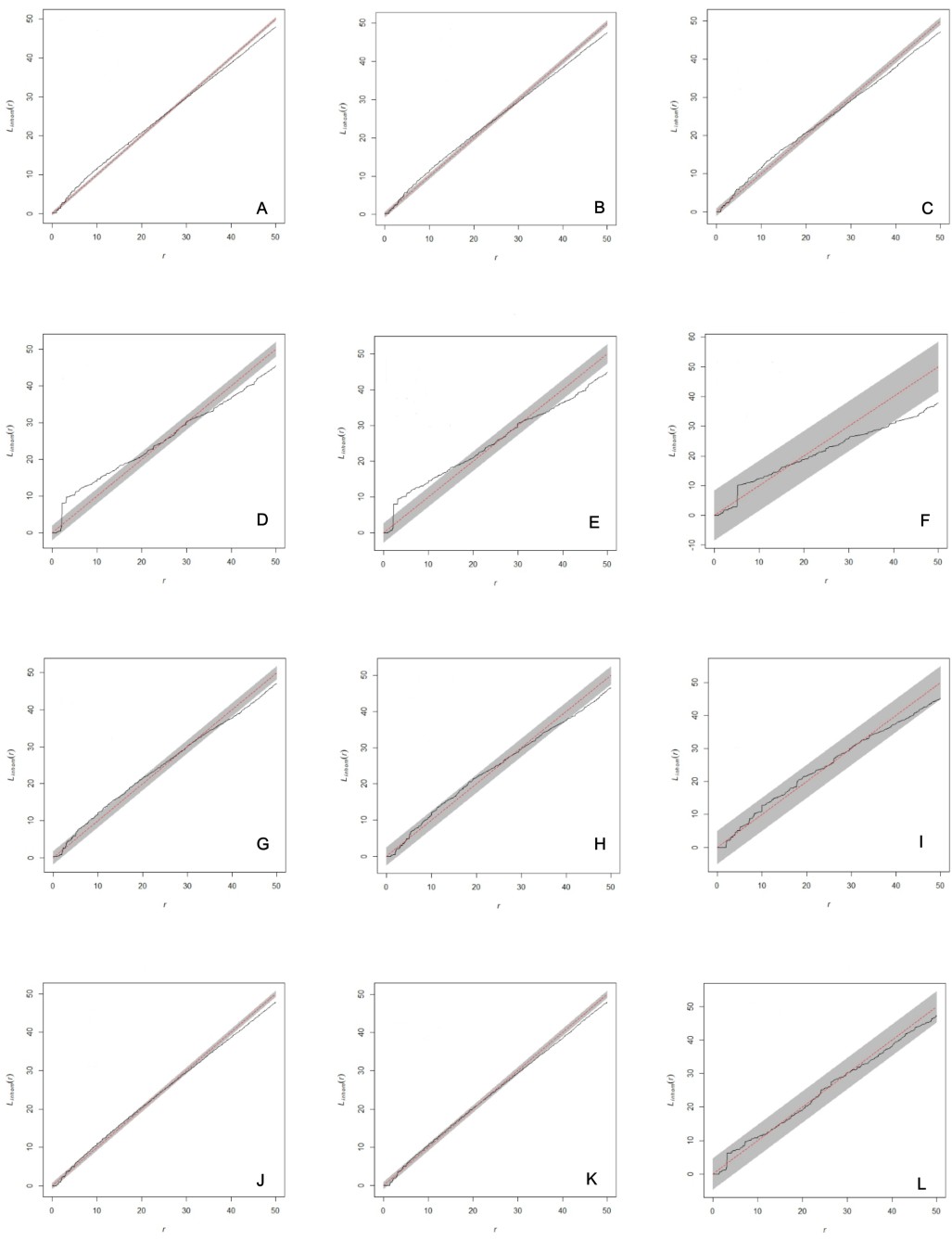

**Figure 2** **Analysis of spatial interaction between all individuals (A, D, G, J), between adults (B, E, H, K), and between females and males only (C, F, I, L) in two populations of *Z. portoricensis*.** El Tamarindo (ET plots) and Susúa Forest (SF plots). (A–C): ET1 plot; (D–F): ET2; (D–F): ET2; (G–I): SF1; (J–L): SF2. In each graph, the solid black line shows test statistic, dashed straight red line shows theoretical values of Linhom(r) for complete spatial randomness, and shaded grey areas show significance envelopes for $\alpha = 0.01$ estimated from 199 replicates of Monte Carlo simulation. For the distance argument ($r$) we used the range recommended by Spatstat (0 to 50 units, 1 unit = 0.1 m).

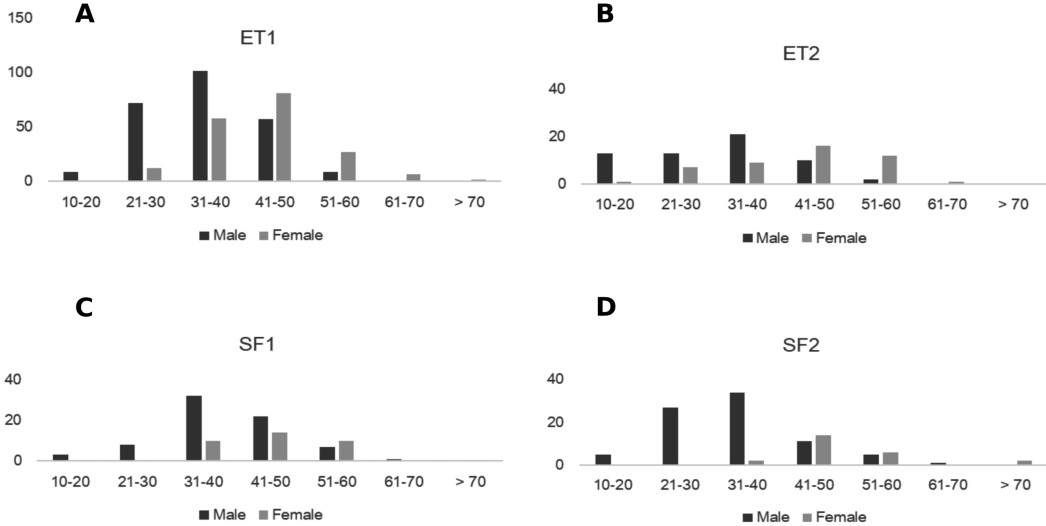

**Figure 3 Size distribution of males and females of *Z. portoricensis* at El Tamarindo (ET) and Susúa State Forest (SF) sites. Size was measured as the number of leaflets on the longest leaf.** (A) ET1 plot, (B) ET2, (C) SF1, and (D) SF2. In each graph, the black bars indicate the number of males at each size class, and the gray bars indicate the number of females at each size class. Size was measured as the number of leaflets on the longest leaf.

small distances (0–1.7 m) where individuals appear clustered and at larger distances, from 3.8 m to 5 m (*r* max), where their distribution is regular (Figs. 2D, 2E); at intermediate distances (1.8–3.8 m) they appear randomly distributed. The set composed by females and males appears randomly distributed at almost all the range of *r* values, except for the 3.8 to 5 m interval where they showed a significant deviation toward regular distribution (Fig. 2F).

## Reproductive effort

Onset of coning was related to plant size and sex. Male plants were reproductively active at sizes below 20 leaflets in the longest leaf and this was similar in all plots. Female plants matured at different sizes (e.g., 24–37 leaflets in the longest leaf) and produced cones at sizes between 21 to 30 leaflets in ET plots, and at sizes above 30 leaflets in SF plots (Fig. 3, Table S1). The average size of male and female plants was statistically different for each plot (Table 2). No significant difference was detected between plots of the same site, but the difference between sites was significant (Table 2). Across all sites, larger females generally produced cones with more ovules, but the relationship was highly variable (linear regression: $N = 33$, $r^2_{adj} = 0.11$, $F_{1,31} = 4.89$, $P = 0.03$).

## Reproductive success

Overall average seed set per cone was 44.7 ± 33.7% and average seeds per cone were 19.0 ± 14.4. Despite differences in plant densities among plots, we failed to detect significant differences among them for either measure of RS (ANOVA on square root transformed data: seed set: $F_{3,59} = 1.51$, $P = 0.22$; seeds: $F_{3,59} = 1.76$, $P = 0.16$; Data S5). Seed set failure occurred in 16% of female cones and half the plants had less than 50% seed set.

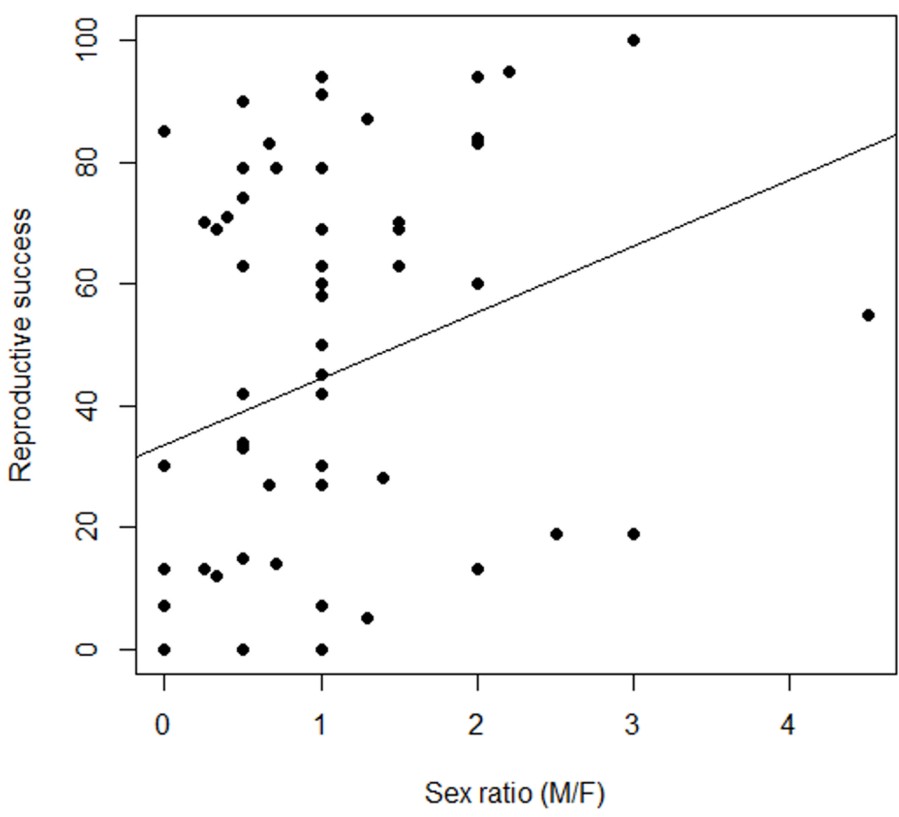

**Figure 4** **Variation in the relationship between reproductive success and sex ratio, males to females (M/F), in Z. portoricensis.** Linear regression: $r^2_{adj} = 0.06$, $F_{1,58} = 4.96$, $P < 0.03$).

**Table 2** **Nested ANOVA analysis of the differences in plant size of Z. portoricensis at three levels: (1) between sites, (2) between plots within sites, and (3) between sexes within plots.**

|  | SS | DF | MS | F | P |
|---|---|---|---|---|---|
| Site | 1,602.0 | 1 | 1,602.0 | 18.488 | <0.001 |
| Plot (Site) | 477.6 | 2 | 238.8 | 2.756 | 0.064193 |
| Sex (Site*Plot) | 15,313.9 | 4 | 3,828.5 | 44.182 | ≪0.001 |
| Error | 64,815.8 | 748 | 86.7 |  |  |

We found no evidence of spatial autocorrelation between position of females and RS in any of the four areas we had sampled (Mantel $r = -0.007$ to $-0.12$, $P = 0.48$ to $0.88$; Data S5)

Female reproductive success was highly variable (Fig. 4; Data S5), but our measures of RS, seed set and number of seeds, were correlated and behaved similarly as response variables. Consequently, we only report seed set as our measure of female RS. There was a non-significant relationship between RS of a target female plant and the number of males within the circular plots ($r^2_{adj} = -0.009$, $F_{1,58} = 0.49$, $P = 0.48$). However, we found a significant positive relationship between female RS and sex ratios of the circular plots

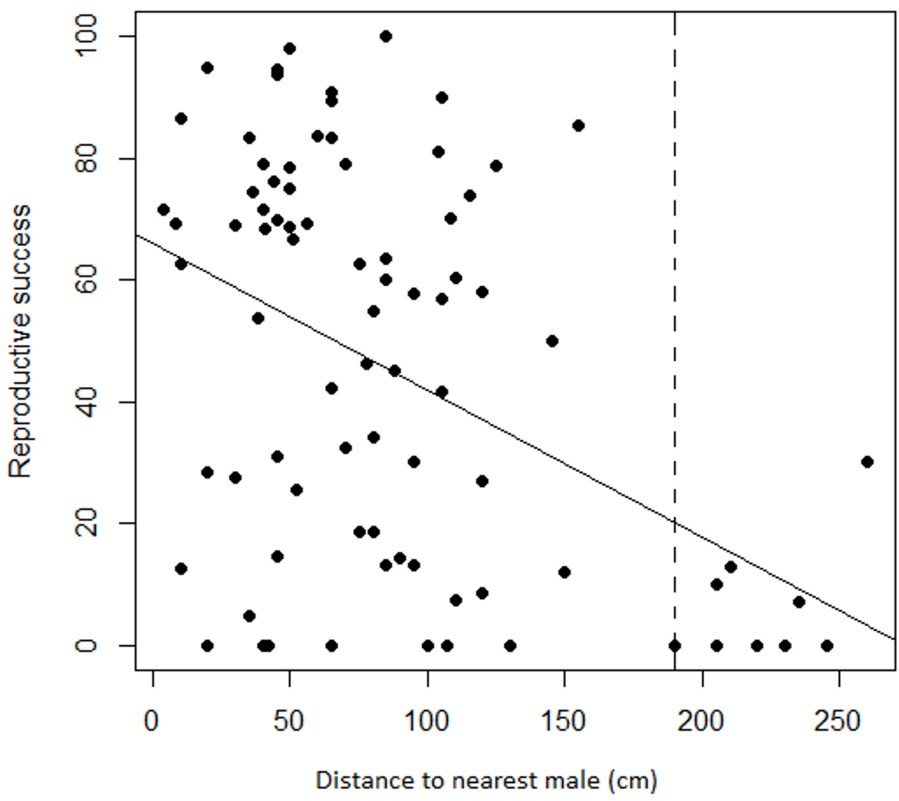

**Figure 5** **Relationship between female reproductive success and distance to the nearest male in *Z. por-toricensis*.** Linear regression: $r^2_{adj} = 0.18$, $F_{1,83} = 19.78$, $P < 0.001$. The dashed line represents a threshold distance (i.e., 190 cm) after which the negative relationship between reproductive success and distance becomes significantly different from zero.

$(r^2_{adj} = 0.06$, $F_{1,58} = 4.96$, $P = 0.03)$. While the slope of line was significantly different from zero, the model explained very little of the variation.

Regression analyses also showed that there was a significant negative relationship between female RS and the distance to the nearest male (adjusted $r^2 = 0.18$, $F_{1,83} = 19.78$, $P < 0.001$). Variation was again substantial but less so at greater distances where seed set was frequently nil. Progressively eliminating the target females with the most distant neighbors changed the relationship from a significant negative one to a slope indistinguishable from zero at a threshold distance of 190 cm (Fig. 5). Only the nine most distant data points were eliminated to reach non-significance, leaving a still substantial sample size of 75.

## Patterns of female success

Seed set (proportion of ovules forming seeds) is significantly and negatively associated with female cone size according to our linear regression analysis ($b = -0.86$, $r^2_{adj} = 0.09$, $F_{1,58} = 6.76$, $P = 0.01$). Thus, larger cones have lower percent seed set, but the model is weak, explaining only 9% of the variation.

Seed set may not reflect fitness advantage to larger cones, but absolute seed production may. We assumed that if a plant has the resources to produce larger cones, then they

would have proportionately greater resources to produce more seed. And larger cones may be better competitors for pollinator services. We certainly expected more seeds from larger cones. Instead, we found that virtually none of the variation in seed production was associated with cone size ($b = -0.03$, $r^2_{adj} = 0.02$, $F_{1,58} = 0.04$, $P = 0.83$).

If intraspecific competition for resources was a significant factor, then high plant densities should have had negative effects on reproductive effort. In our circular plots, we found no such correlation with female cone size (Spearman's rank correlation: $Rho = 0.11$, $P = 0.39$).

## DISCUSSION

### Population characterization

The population structure of cycads, as in many long-lived perennials, is generally characterized by an "Inverted J" shape, with large numbers of individuals at seedling and early juvenile size classes and very few individuals in the late adulthood size class (*Negrón-Ortiz & Breckon, 1989b*; *Vovides, 1990*; *Watkinson & Powell, 1997*; *Pérez-Farrera et al., 2000*; *Pérez-Farrera et al., 2006*; *Lazcano-Lara, 2004*; *Yáñez Espinosa & Sosa-Sosa, 2007*; *López-Gallego, 2008*; *Gerlach, 2012*; *Álvarez Yépiz, Búrquez & Dovčiak, 2014*). With 50–80% of the individuals in our populations being seedlings and juveniles, the structure may not be as skewed as in other cycad populations, but it certainly suggests that plants are thriving in both El Tamarindo and Susúa State Forest populations, particularly compared to most populations of *Z. portoricensis* where recruitment is very limited or absent (*Lazcano Lara, 2015*). Although cycads are long-lived plants that exhibit long-term reproductive potential and may use asexual regeneration to persist, successful recruitment is critical for the long-term viability of the populations (*Raimondo & Donaldson, 2003*; *Álvarez Yépiz, Dovčiak & Búrquez, 2011*).

Biased operational sex ratios are common in populations of dioecious species despite theoretical expectations of an equilibrium 1:1 rato (*Field, Pickup & Barrett, 2012*), and the populations of *Z. portoricensis* we studied are no exception. Three of our four plots were significantly male-biased. However, in cycads, male-biased operational sex ratios seem to be as frequent as equal sex ratios. Populations with more males than females have been reported in nine cycad species (*Ornduff, 1985*; *Ornduff, 1987*; *Clark & Clark, 1987*; *Tang, 1990*; *Negrón-Ortiz & Gorchov, 2000*; *Pérez-Farrera et al., 2000*; *Hall et al., 2004*; *Kono & Tobe, 2007*; *Proches & Johnson, 2009*), whereas statistically equal proportions of males and females have been reported for another seven species (*Ornduff, 1987*; *Watkinson & Powell, 1997*; *Raimondo & Donaldson, 2003*; *Lazcano-Lara, 2004*; *Pérez-Farrera et al., 2006*; *Yáñez Espinosa & Sosa-Sosa, 2007*). Thus far, female biased operational sex ratios are rare, having been reported in only two species (*Newell, 1983*; *Mora et al., 2013*).

Females generally have a higher cost of reproduction than males (*Obeso, 2002*). Differential reproductive investment, particularly in long-lived species, has been proposed as the main reason for operational male-biased sex ratios (*Field, Pickup & Barrett, 2012*). Indeed, *Clark & Clark (1987)* and *Tang (1990)* demonstrated that female cycads invest more in reproduction than do males and may not have the resources to cone every year.

However, an increase in habitat quality, mainly in light availability, may compensate for the differences between sexes in reproductive investment (*Clark & Clark, 1987*), so that more females produce cones (i.e., fewer non-reproductive adult females) resulting in a 1:1 operational sex ratio as *Ornduff (1987)* found in *Z. pumila* L. In the present study, habitat quality also may have affected sex ratios. The difference in operational sex ratios was less pronounced at El Tamarindo plots (Table 1) where plants received more light than those at the Susúa State Forest (approximately 28% difference in canopy coverage). On the other hand, the greater photosynthetic capacities for *Z. portoricensis* males reported by *Krieg et al. (2017)* may have influenced the more male-dominated operational sex ratios in the low light Tamarindo plots. Furthermore, water conservation could play a role as well since *Krieg et al. (2017)* found that female *Z. portoricensis* has greater water-use-efficiency than do males. Unfortunately, these physiological performance traits were not linked with costs of reproduction or coning behavior.

## Spatial patterns and interactions between individuals

Limited dispersal and environmental heterogeneity are expected to result in an aggregated spatial arrangement of individuals, but in these cases clustering does not necessarily imply dependence or attraction between them (*Law et al., 2009*). Aggregated distribution of individuals in cycad populations occurs in *Dioon edule* (*Vovides, 1990*), *Ceratozamia matudai* (*Pérez-Farrera et al., 2000*), *Ceratozamia mirandae* (*Pérez-Farrera et al., 2006*), *Macrozamia ridlei* (*Gerlach, 2012*), and *Dioon sonorense* (*Álvarez Yépiz, Búrquez & Dovčiak, 2014*). Limited dispersal and habitat heterogeneity were hypothesized as major factors influencing the spatial pattern, although, in all the cases, animal dispersal of a portion of the seeds was suspected (*Vovides, 1990*; *Pérez-Farrera et al., 2000*; *Pérez-Farrera et al., 2006*; *Hall & Walter, 2013*) or corroborated (*Gerlach, 2012*). *Vovides (1990)* hypothesized that competition among *D. edule* and angiosperms was the main cause of the aggregated pattern, as rapid growing angiosperms were using spaces with deep soil layer, confining cycads to patches with thin soil layers and rock surfaces. Indeed, *Álvarez Yépiz, Búrquez & Dovčiak (2014)* found that interspecific competition with angiosperms in a semi-arid habitat is associated with lower growth rates for adults in sparse populations of *D. sonorense*. While we did not specifically test for interspecific competition, it has not generated obvious spatial patterns of this the dominant understory element (Fig. 2, Fig. S1).

The spatial pattern exhibited by *Z. portoricensis*, non-overdispersed individuals at distances between 0–3.8 m, but without any interaction between them (Fig. 2), has two notable implications for the reproductive biology of the species in these populations. First, proximity among individuals might provide stability for a deception-based pollination system with rewardless female plants, since pollinators will have an increased probability of finding a reward (i.e., a male plant with an available cone) after being deceived which reduces the cost for the insects and may make it unnecessary to increase their ability to identify deceptive females (*Renner, 2006*). Secondly, the lack of dependence among individuals means that the presence of both sexes in the same place is random; thus, access to mates and, consequently, RS will be subjected to high levels of stochasticity. Indeed, we have revealed high variation in RS in all our plots.

## Reproductive success

Auto-deception pollination in cycads and other plants is dependent on a balance between competing needs of plants and pollinators. The reward obtained by the pollinators from using male sexual structures, is countered by the cost of visiting deceitful, non-rewarding female sexual structures. The plant's gain in deceit is likely ovule escape from herbivory, but the cost may be pollen limitation (*Norstog & Nicholls, 1997*; *Dufaÿ & Anstett, 2003*). This mutual exploitation appears to be sufficiently stable for long-term population persistence, even when in various cycad species seed set values rarely go over 60% with many plants showing evidence of receiving few or no pollen at all (*Newell, 1983*; *Tang, 1987a*; *Kono & Tobe, 2007*; *Lazcano Lara, 2015*; *Proches & Johnson, 2009*; *Suinyuy, Donaldson & Johnson, 2009*; *Terry, 2001*; *Wilson, 2002*). Half the plants assessed in this study had seed set below 50%, and 16% of the plants did not produce any seeds.

Resource constraints clearly affect the initiation of cone development in this and other species (*Clark & Clark, 1987*; *Tang, 1990*), but the degree by which it affects seed set is not clear. While female cone size was weakly but significantly related to plant size, expected outcomes of resource limitations to female cone size, seed set and seed production were wanting, despite relatively high plant densities. On the other hand, our indirect evidence was consistent with expectations of pollen limitation as seen in other deception pollinated plants, including dioecious species with male biased sex ratios (*Agren, Elmqvist & Tunlid, 1986*; *Tremblay et al., 2005*). More definitive evidence is needed, but is difficult to obtain largely because of inherent difficulties in ascertaining the efficacy of hand pollinations (e.g., are undeveloped ovules the result of fertilization failure or pollen puffed into cones failing to contact pollination droplets?).

Pollination success in cycads, as in other zoophilous dioecious plants, is dependent on the simultaneous presence of pollen producing males within the flight capabilities of their pollinators. If zamias are pollen limited, then receptive females are competing for pollination service, which may be influenced by the spatial arrangement of females to males. Pollen availability should be higher for females residing in neighborhoods dominated by males. Indeed, we found a significant positive relationship between female RS and the immediate-neighborhood operational sex ratio whereby plants did better where there were proportionately more males (Fig. 4). The abundance of males around a female was less important than the distance to the nearest male. However, reproductive success was highly variable, suggesting that either a high stochastic element to the probability of pollinator visits exists or that other factors may be involved, such as variation in traits that attract pollinators to female cones.

In deceptive pollination systems involving automimicry in dioecious plants, pollination is a collateral benefit for female plants because insects do not profit from visiting female sexual structures (*Little, 1983*; *Willson & Agren, 1989*; *Dufaÿ & Anstett, 2003*). In cycads, females mimic the fragrance profile and thermogenesis pattern of males but this mimicry is not perfect, and females emit odors in lower amounts, with small variations in the volatile compounds (*Pellmyr et al., 1991*; *Terry et al., 2004*; *Donaldson, 2007*; *Proches & Johnson, 2009*; *Suinyuy, Donaldson & Johnson, 2010*). As in other model-mimicry systems, the closer the mimic is to the model, the better the reproductive success (*Peter & Johnson, 2008*). In

*Z. portoricensis*, the probability of female plants being detected by beetles should be higher the closer they are to male plants. At the same time, proximity to a male should reduce the cost of visiting a rewardless female for the deceived insects. *Proches & Johnson (2009)*, in their study of *Stangeria eriopus* pollination, found no effect of distance to the nearest male on fecundity ($r^2 = 0.043$, $df = 61$, $P = 0.114$), whereas we found that seed set in *Z. portoricensis* is significantly associated with the proximity to males, revealing mate-finding Allee effects when distances to males reached 1.9 m and beyond. Why do the two cycads differ in this regard? We propose that the Nitidulidae beetle pollinators of *Stangeria eriopus* are more free-ranging because they are neither cycad specialists nor do they use male cones as a brood place. They are generalists that usually consume fermented fruit and are attracted by the fermented fruit odors produced by both male and female *S. eriopus* cones (*Proches & Johnson, 2009*). For cycads whose pollinators use cones as brood sites, pollen dispersal distances may be more limited. Alternatively, gene flow distances may simply be density dependent. Where population densities are high, gene flow is limited (e.g., *Duminil et al., 2016*), and this may account for differences among cycad populations.

*Pellmyr et al. (1991)* showed that odor emissions in *Zamia* are weak compared to other cycad genera. This may explain why, in the absence of strong signaling mechanisms, mate proximity is more critical for the pollination of *Z. portoricensis* than for other cycads. However, the relative low strength ($r^2_{adj} = 0.18$) of this relationship suggests that either a high level of stochasticity exists or other factors in addition to mate proximity affect seed set in *Z. portoricensis*. For example, pollination will not occur if nearby male plants fail to attract and host pollinators, or if receptivity of female cones is asynchronous with pollen release by male cones.

## Conservation applications

Cycads are considered "living fossils", and indeed the major lineages are old (*Chamberlain, 1919*; *Chamberlain, 1935*; *Norstog & Nicholls, 1997*). Extant species, though, are not relicts, having diverged within the last 19 my (*Condamine et al., 2015*) suggesting that the lineage has had some degree of evolutionary flexibility in relatively recent times. Yet habitat destruction and collecting have endangered many of these species throughout the world (*Donaldson, 2003*). Some may be more resilient to habitat perturbations than others. Forest coverage in Puerto Rico had been reduced to less than 5% by the 1940s (*Roberts, 1942*; *Wadsworth, 1950*), but recovery has progressed ever since with forests now covering approximately 40% of the island (*Grau et al., 2003*; *Kennaway & Helmer, 2007*; *López Marrero & Villanueva Colón, 2006*). Remarkably, *Z. erosa* has become abundant and populations are stable (*Negrón-Ortiz, Gorchov & Breckon, 1996*); *Z. portoricensis* thrives locally at Susúa State Forest, and surrounding areas in Sabana Grande and Yauco municipalities but elsewhere occurs in populations where recruitment is very limited or absent; and *Z. pumila* L. occurs in small populations with little or no pollinator activity (*Lazcano Lara, 2015*). Active management involving introductions, reintroductions, and translocations is becoming a necessary complement to habitat protection. Our results underscore the need for management plans to incorporate spatial aspects that will facilitate effective access to mates, and with an adequate number of plants of both sexes to maximize

reproductive success, and promote self-sustainable, viable populations of both plants and their specialist pollinators.

## CONCLUSIONS

*Z. portoricensis* is well established at the eastern portion of the southern serpentine outcrop of Puerto Rico. Its abundance in those places is among the highest reported for a cycad, and its population structure suggests regular successful recruitment.

Reproductive success of *Z. portoricensis* at the individual level is affected by the spatial distribution of the plants. Although reproductive success is highly variable, as a trend, it decreases with an increase in the distance to the nearest male, a relationship that becomes statistically significant at distances beyond 190 cm. Reproductive success also tends to be higher for female plants in neighborhoods with more coning males than females. As the individuals are randomly located in relation to each other, being close to male plants and/or located within male dominated sites is a matter of chance. The effect of this stochastic process on reproductive success is perhaps one of the factors that produce the high levels of intra-population genetic diversity found by *Meerow et al. (2012)* in this species.

The remarkable long-term persistence of the cycads indicates the effectiveness of their sequential nursery-deception pollination system. The variability of its effectiveness at the individual level in *Z. portoricensis* could be interpreted as the result of the stochasticity associated with random spatial distribution of individuals of both sexes, infrequent pollinator visits to females and the lack of synchronization between pollen release and female receptivity. But cycads are long-lived plants so net reproductive success is perhaps best assessed via long-term studies (i.e., 20 years or more) to capture fluctuations in fitness.

## ACKNOWLEDGEMENTS

The authors thank JK Zimmerman for many useful recommendations. María E. Ocasio-Torres provided valuable assistance during the fieldwork. Christopher Krieg and two anonymous reviewers provided valuable critiques of our work.

### Funding

Travel was partially funded by the Center for Applied Tropical Ecology and Conservation and the NSF-CREST program (HRD-0734826, Elvira Cuevas, Project Director). There was no additional external funding received for this study. The funders had no role in study design, data collection and analysis, decision to publish, or preparation of the manuscript.

### Grant Disclosures

The following grant information was disclosed by the authors:
Center for Applied Tropical Ecology and Conservation.
NSF-CREST program: HRD-0734826.

## Competing Interests

The authors declare there are no competing interests.

## Author Contributions

- Julio C. Lazcano-Lara conceived and designed the experiments, performed the experiments, analyzed the data, contributed reagents/materials/analysis tools, prepared figures and/or tables, authored or reviewed drafts of the paper, approved the final draft.
- James D. Ackerman conceived and designed the experiments, analyzed the data, contributed reagents/materials/analysis tools, authored or reviewed drafts of the paper, approved the final draft.

## Field Study Permissions

The following information was supplied relating to field study approvals (i.e., approving body and any reference numbers):

The Departamento de Rercursos Naturales y Ambientales de Puerto Rico approved this study (DRNA 2012-IC-028, DRNA 2013-IC-022, and DRNA 2014-IC-033).

## Data Availability

The raw data are provided in the Supplemental Files.

## Supplemental Information

Supplemental information for this article can be found online at http://dx.doi.org/10.7717/peerj.5252#supplemental-information.

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
