# Peer review of "Best in the company of nearby males: female success in the threatened cycad, Zamia portoricensis"

_PeerJ, doi:10.7717/peerj.5252_

## Round 0.1 · original submission · Minor Revisions

When revising your manuscript, please take into consideration all remarks made by reviewers, with particular reference to the suggestions provided by Rev#2 who requires a stronger connection to the conservation implications throughout this manuscript.

Reviewer 1 ·

Basic reporting

- Clear and unambiguous, professional English used throughout.
- Literature references: very good, except regarding the Allee effect mentioned in the text. The points in the text regarding the Allee effect should be modified, explaining better what is the Allee effect (shortly) and how it is related to this study (more deep). Gascoigne et al. 2009 cited (be careful at line 472 you wrote GASCOIGEN) is about mate-finding Allee effects, and I strongly suggest you to add 10 lines in the Introduction section to introduce the concept of mate-finding Allee effects and how it is crucial for the comprehension of several aspects of the reproductive ecology (and success) of vascular plants.
- Professional article structure, figs, tables are OK. Raw data shared: YES
- Self-contained with relevant results to hypotheses: YES (but see below for aims)

Experimental design

- Original primary research within Aims and Scope of the journal: YES
- Research question well defined, relevant & meaningful: I have some doubt regarding aim #2: "is pollen availability associated with reproductive success?" Indeed you did not measure pollen production, vitality and availability in this study. As you state at line 159 "Pollen availability was assumed to increase with an increase in the male-to-female sex ratio". At line 181 you write: "reproductive success is assumed to be pollen limited". There are too much assumptions related to pollen production, that should be carefully discussed somewhere in the text, and discourage the use of "pollen availability" in the aims. So in the aims, you better use another term, not mentioning pollen availability: a reader would expect to find an analysis of pollen production and/or pollination efficiency.
- Rigorous investigation performed to a high technical & ethical standard: YES
- Methods described with sufficient detail & information to replicate: YES

Validity of the findings

- Impact and novelty of the work: YES
- Data is robust, statistically sound, & controlled: this is the weakest point of the manuscript: Authors declare to have studied two populations of Z. portoricensis (line 88) and to have established two 25x20 m plots at each site. Even if I understand that there could be many logistic difficulties to expand the range and number of surveyed populations, I think that somewhere in the text Authors should discuss briefly the weakness of studying "only" 4 plots (on a total studied area of "only" 1000 square meters). Furthermore, as they write at line 396, Authors worked at places characterized by high (or very high) densities for Cycads, whereas it is demonstrated that different ecological processes affect plant populations at their range limits, where they usually (but not always) show low densities (see, among others: Pironon et al. 2017. Biological Reviews 92: 1877–1909; Papuga et al. 2018. Ecography 40: 1-15). So, to my opinion, not only this study has the limit of a small spatial range and replicates, but its results and main conclusions can be valid only at sites with high cycads density, not where cycads have low densities. I recognize that it is probably not possible to expand the replicates and/or sites of the study in a short period (to improve the manuscript), but clarifications should be included in the manuscript, about the limits of the study and the applicability of the results here presented.
- Conclusion are well stated, linked to original research question & limited to supporting results: YES, but see comment above

Additional comments

Congratulations for this interesting, even if spatially limited, research.
I suggest you also to stress somewhere that you are dealing with probably the oldest seed plant group insect pollinated, this will increase the originality of your paper: as being the "first" Gymnosperms insect pollinated, Cycads were possibly the first seed plants experimenting also the ecological effects of pollen/pollination/pollinators limitation, constraints not experienced by other Gymnosperms like conifers.

·

Basic reporting

Title; I suggest revising the title to be more descriptive and with no innuendo.
Abstract; Please use more conservative language when statements are not supported by data. For example, please change “…indicates that pollinator movement among plants is limited for this mutually…” to include “may be limited”, given that there are no data presented on pollinator movement.
Line 36-39; The first sentence is very busy and has a vague conclusion (“…influences the effects that different ecological factors have…”). I recommend splitting this sentence up and being more specific.
Line 42; The sentence starting with “For example..” does not exemplify the prior sentence’s subject ie genetic diversity. I suggest revising the paragraph to improve the logical flow.
Line 48-56; This paragraph seems like an appropriate place to mention dispersal mechanisms. The papers Burbidge & Whalen 1982, and Hall & Walter 2013 come to mind. Please mention dispersal mechanisms briefly as it relates strongly to your study.
Line 52-54; This sentence reads disrupted and could be more clear. I suggest revising to something like “Cycads may be particularly susceptible to Allee effects given that they are dioecious, and the majority of cycad species are threatened with extinction and have narrow distributions.” In my opinion, the goal here would be to connect reproductive strategy (dioecy) and population ecology (via invoking threatened status and implicitly from "narrow distributions").
Line 64-86; A few studies have shown that male and female reproductive cycles are often offset and that although there may be reproductive females each year, individual females often skip years. See work by Clark & Clark on Zamia sp. So here, I recommend citing such studies and comparing the observed reproductive trends of this species at these sites specifically.
Line 140; There has been some limited work in cycad species showing data to support the hypothesis that male and female cycads differ in their cost of reproduction but the only physiological study to date is Krieg et al 2017. This study included Z. portoricensis and showed that males and females differ in key aspects of physiology. This work should be cited here, and probably in the discussion as well.
Line 161; Typo – please add : between “variables” and “number of males”
Line 195; Please add a space between “reproductives,” and “to”
Line 260-267; Another place for Alverez-Yepiz et al 2014
Line 272; Citation needed after “…equal sex ratios”
Line 298; Consider adding Burbidge & Whalen 1982, and Hall & Walter 2013
Line 379; Please clarify what you mean by “flexibility”.

Experimental design

Line 96 & line 99; In regards to the percentages reported, please add a metric of variation, eg standard deviation.
Line 118-119; Given that at least three platforms were used for analyses (eg R, Statistica, JMP), I suggest being more specific about what which analysis was performed in each. This should be implemented throughout. Please specify which platform was used for each analysis.
Line 139; This may be true in other plant systems but there is much less data for cycads. Please add some rational and citation to support this expectation.
Line 141; Why did you choose the number of leaflets to rank whole plants into size classes? Please add some rationale and citations to support this decision.
Line 142-143; This expectation that female cone production will me dependent of plant size should either be well-supported with citations and the findings of other studies, or perhaps made a hypothesis.
Line 157-158; It seems like the absolute number of males is a far more direct approach to estimating the availability of pollen than the sex ratio. Imagine the scenario where two plots have a 50/50 sex ratio. One plot has one male and one female and the other plot has 10 males and 10 females. Given that pollen is made in ‘surplus’, the relationship between pollen limitation (in females) and number of males is not 1:1, but what I would expect to be a quickly saturating curve where a few males could theoretically supply an exponentially high number of females. Please also analyze your data with number of males or density of males.
Line 160; I do not think your assumption is valid. Please see above comment and re-run analyses with number or density of males.
Line 177-178; Was the threshold different across sites?
Line 180; This section isn't about multiple resources, it only addresses pollen. I suggest renaming the section to something like: Pollen Limitations, Assessing the role of pollen availability, etc
Line 213; Is it 4-5 meters or 3.8 to 5 meters as stated in line 210. Please clarify.
Line 220 & 222; My personal preference is to see the corresponding statistics in parentheses after stating the results.
Line 226-227; Is the standard deviation also in percent? Please make units consistent and clear.
Line 234; Please remove the word ”highly” due to the low predictive value (ie 18%).
Line 236; Please remove the word “positive” due to the statistical results indicating the slope is not significantly different from zero.

Validity of the findings

Line 252-253; I am having trouble seeing the connection between the assertion that “If pollen is not limiting seed production, then we would expect more seeds produced in larger cones”, and I do not think this assertion is logical. The second part of the assertion that larger individuals may produce more seeds is introduced earlier in the manuscript and without support or citations. Here, you have linked pollen limitation, again without support, and have not explained your logic for this connection. I strongly recommend revision of these assertions to include rationale and supporting literature.
Line 254-256; To my knowledge, this assertion about intra-specific competition leading to reduced reproduction is untested in cycads. The closest study I know is Alverez-Yepiz et al 2014, who showed that survival and grow rates both decreases in a Dioon sp with increasing intraspecific competition (ie density). Please situate your assertion within other studies including Alverez-Yepiz et al.
Line 266-267; A large proportion of seedling and juveniles relative to the number of adults does not suggest a thriving species; it may suggest the opposite. Please expand your discussion of seedling survival and why you think your sites offer favourable conditions.
Line 272-275; There have been many abiotic explanations proposed, some with data, to explain sex biased populations. Please cite and discuss.
Lines 280-291; This paragraph should include Krieg et al 2017 who showed that males of Z. portoricensis have higher photosynthetic rates than females, but that females of this species have greater water-use-efficiency – the opposite trend one could expected if females face greater costs in reproduction. Even so, females may show longer reproductive cycles because they play a more “conservative” game with reduced photosynthetic capacity and more conservative stomatal control. Please cite and discuss.
Line 301-303; Please add Alverez-Yepiz 2014 to support your reasoning and discuss alternative hypotheses from Burbidge & Whalen 1982, and Hall & Walter 2013
Line 303-306; The statement about interspecific competition is misleading here. Just because a species is dominant doesn’t not mean that interspecific competition is not occurring. Further, the sentence posits that interspecific competition is only “important” when the study species is the “loser”. Without data on interspecific competition, all you can say here is that “interspecific competition does not appear to be excluding Z. portoricensis from these sites since individuals of Z. port comprise a dominant understory element.
Line 363-366; You offer an interesting idea to explain the discrepancy between studies but stop short of an actual hypothesis, leaving this paragraph with something of a cliff-hanger. Please elaborate on your proposed connection between specialist vs generalist impacts on the relationship between RS and sex ratios/proximity to the opposite sex.
Line 373; I am not a horticulturalist but I understand asynchronous reproductive cycles between males and females to be common in many cycad species. Please use more conservative language and add revise to “…female cones is too asynchronous…”
Line 377; Please add questions around the popular phrase “living fossils” due to the oxymoronic nature of the name and the fact that it is technically inaccurate.
Line 378-379; Salas-Leiva et al., 2013 is not the foremost appropriate study to cite here. Nagalingum et al 2011 proposes that extant cycad species are as young as 12 myo, but this is the minimum age of some taxa and others are very clearly older. Moreover, subsequent phylogenetic studies from the same group have acknowledged some issued and violated assumptions with that 2011 study, eg Condamine et al 2015. Please add these studies and revise your statement to reflect the conclusions of these studies, ie that extant taxa are 20+ myo.

Additional comments

Overall, this is an interesting study and addresses fundamental questions of cycad biology that remain poorly understood. In particular, there have been few studies that examine cycad reproduction in natural populations. This study fills this gap and makes important connections to conservation efforts.
Generally, I recommend a stronger connection to the conservation implications throughout this manuscript. The conservation implications of this work are exciting and far better situated in the data presented than the attempts to make conclusions about pollinators. In some cases, the focus on pollinators manifested as unsupported conclusions and assertions for which no data are presented. I strongly recommend a thorough review of all language and statements relating your data to pollinator interactions to a more conservative stance on the implications.

Reviewer 3 ·

Basic reporting

No comment.

Experimental design

No comment.

Validity of the findings

No comment.

Additional comments

This is well-written, well-structured contribution that does an excellent job of hypothesis testing without over-stating its conclusions. It makes a useful contribution to the area of cycad reproductive biology. See some of my comments in the PDF.

Annotated reviews are not available for download in order to protect the identity of reviewers who chose to remain anonymous.

---

## Round 0.2 · accepted · Accept

I acknowledge the considerable effort you made in replying to all comments in details, including supporting literature for explaining your choices. As far as the title is concern, the reviewer is probably right when he/she is asking a more descriptive one, but finally I agree with you that your solution is more intriguing and recalls the real nature of your work.

#